# Upregulation of MDR- and EMT-Related Molecules in Cisplatin-Resistant Human Oral Squamous Cell Carcinoma Cell Lines

**DOI:** 10.3390/ijms20123034

**Published:** 2019-06-21

**Authors:** Hyeong Sim Choi, Young-Kyun Kim, Pil-Young Yun

**Affiliations:** Department of Oral and Maxillofacial Surgery, Section of Dentistry, Seoul National University Bundang Hospital, 82 Gumi-ro 173 beon-gil, Bundang-gu, Seongnam-si, Gyeonggi-do 13620, Korea; simmi0326@gmail.com (H.S.C.); kyk0505@snubh.org (Y.-K.K.)

**Keywords:** OSCC, Cisplatin, BCRP, MDR1, EMT

## Abstract

Cisplatin is one of the major drugs used in oral cancer treatments, but its usage can be limited by acquired drug resistance. In this study, we established three cisplatin-resistant oral squamous cell carcinoma (OSCC) cell lines and characterized them using cell viability assays, qPCR, Western blotting, FACS, immunofluorescence, and wound healing assays. Three OSCC cell lines (YD-8, YD-9, and YD-38) underwent long-term exposure to cisplatin, eventually acquiring resistance to the drug, which was confirmed by an MTT assay. In these three newly established cell lines (YD-8/CIS, YD-9/CIS, and YD-38/CIS), overexpression of multidrug resistance (MDR)-related genes was detected by qPCR and Western blotting. The cell lines displayed an increase in the functional activities of breast cancer resistance protein (BCRP) and multidrug resistance protein1 (MDR1) by rhodamine 123 and bodipy FL prazosin accumulation assays. Moreover, the cisplatin-resistant cells underwent morphological changes, from round to spindle-shaped, increased expression of epithelial-to-mesenchymal transition (EMT)-related molecules such as N-cadherin, and showed increased cell migration when compared with the parental cell lines. These results suggest that these newly established cell lines have acquired drug resistance and EMT induction.

## 1. Introduction

Oral cancer is an increasingly global disease [1,2], however, more than half of these cancers are found late, sometimes permanently altering a patient’s ability to chew, swallow, talk, as well as their appearance [3]. Therefore, it is important to treat oral cancers at the early stages with minor surgery. Overall the treatment requires an efficient combination of surgery, radiotherapy, and chemotherapy [4,5]. 

Cisplatin, a platinum-based drug, is one of the standard first-line chemotherapeutic agents used to treat OSCC [6]. Unfortunately, several factors can lead to the failure of chemotherapy treatments, including acquired resistance to the chemotherapeutic agents [7,8]. Some research has shown that acquired chemoresistance is associated with overexpression of the ATP-binding cassette (ABC) transporters, which pump drugs out of cells [9,10]. There are 48 known ABC transporters in humans, including MRP1-6, MRP9, MDR1, and BCRP [10,11]. 

Epithelial-to-mesenchymal transition (EMT) is a process by which epithelial cells lose their cell polarity and cell–cell adhesion, and gain mobility and invasiveness to become mesenchymal stem cells [12]. In most cases, EMT is characterized by a loss of E-cadherin, an increase in N-cadherin, and a change in cell morphology from round to spindle-shaped cells [12,13]. A strong correlation has been found between EMT and drug resistance [13,14,15]. 

The purpose of our study was to establish three cisplatin-resistant human OSCC cell lines, and evaluate their biological characteristics compared to their respective parental cell lines.

## 2. Results

### 2.1. YD-8/CIS, YD-9/CIS, and YD-38/CIS Cells Acquired Resistance to Cisplatin, and Only YD-9/CIS Cells Displayed Cross-Resistance to Paclitaxel and CKD-602

Three OSCC cell lines, YD-8, YD-9, and YD-38, were continuously exposed to stepwise increasing concentrations of cisplatin over a one-year period (final cisplatin concentration of 2 μg/mL). An MTT assay was performed to determine whether the cell lines had developed cisplatin-resistance. Figure 1 shows that YD-8/CIS, YD-9/CIS, and YD-38/CIS cells were less sensitive to cisplatin in a dose-dependent manner than their respective parental cells. The cells were also tested for sensitivity to paclitaxel, CKD-602, and 5-fluorouracil. While YD-9/CIS cells were less sensitive to paclitaxel and CKD-602 than YD-9 cells, cell viability did not differ significantly between the other paired cell lines with these treatments. 5-fluorouracil treatments did not lead to differences in cell proliferation between the parental and cisplatin-resistant cells. These results confirm that long-term exposure to cisplatin in YD-8, YD-9, and YD-38 cells led to cisplatin-tolerance. Moreover, YD-9/CIS cells showed cross-resistance to paclitaxel and CKD-602 at higher drug concentrations (0.005, 0.01 μg/mL and 0.1 μg/mL respectively).

### 2.2. MDR-Related Gene Expression Was Altered in the Cisplatin-Resistant Cell Lines. MDR1 Expression Increased in YD-8/CIS and YD-9/CIS Cells, and BCRP Levels Increased in YD-8/CIS and YD-38/CIS Cells 

Next, we investigated the expression of MDR-related genes in both the parental and cisplatin-resistant cell lines. The mRNA expression of MDR-related genes was investigated by qPCR (Figure 2a). In YD-8/CIS cells, *MDR1* (>2.4-fold change), *BCRP* (>7.2-fold change) and *MRP3* (>2.5-fold change) were upregulated. In YD-9/CIS cells, *MDR1* (>7.5-fold change), *MRP4* (>2.7-fold change), and *BCRP* (>1.6-fold change) were upregulated. In YD-38/CIS cells, *BCRP* (>3.7-fold change) was upregulated. We confirmed the expression changes in *BCRP* and *MDR1* by Western blot (Figure 2b). BCRP was upregulated in YD-8/CIS and YD-38/CIS cells, while MDR1 was downregulated only in YD-38/CIS cells. Conversely, MDR1 was overexpressed in YD-8/CIS and YD-9/CIS cells. Our results show that compared with their parental cells, BCRP was upregulated in YD-8/CIS and YD-38/CIS cells, while MDR1 was upregulated in YD-9/CIS cells.

### 2.3. Acquisition of Resistance to Cisplatin Increases MDR1 Activity in YD-9/CIS Cells and BCRP Activity in YD-8/CIS, YD-9/CIS, and YD-38/CIS Cells

We also examined MDR1 protein activity using rhodamine 123, and BCRP protein activity using bodipy FL prazosin. Figure 3a shows that the accumulation of rhodamine 123 was reduced in YD-9/CIS cells (34.4%) compared to parental cells (54.2%). In the three cisplatin-resistant cell lines, the intracellular accumulation of bodipy FL prazosin was less than in their parental cell lines (Figure 3b; YD-8/CIS, 17.5%; YD-8, 65.0%; YD-9/CIS, 25.8%; YD-9, 66.0%; YD-38/CIS, 52.0%; YD-38, 78.2%). These results show that YD-8, YD-9, and YD-38 cells developed cisplatin-resistance by increasing the activity of MDR1 or BCRP.

### 2.4. Cell Lines Which Acquired Cisplatin-Resistance Displayed Increased EMT-Related Markers, Including Cell Mobility, Increased N-Cadherin Expression, and Decreased E-Cadherin Expression 

It has been reported that the acquisition of chemoresistance is associated with an EMT phenotype. We assessed this in our cell lines by immunofluorescence, Western blot, and wound healing assays. Figure 4a,b shows morphological changes between the cisplatin-resistant cells and their parental cell lines, going from round to spindle-shaped cells. N-cadherin expression was increased, while E-cadherin expression was decreased in the cisplatin-resistant cells compared to each parental cell line. Moreover, acquired cisplatin resistance increased the number of cells that migrated into a wounded area in OSCC cell lines (Figure 4c). These data indicate that cisplatin resistance caused molecular changes consistent with a cell migration and EMT phenotype, including modifying E-cadherin and N-cadherin expression.

## 3. Discussion

Cisplatin, a platinum-based drug, is one of the most commonly used chemotherapeutic agents for oral cancers, however, its application can lead to drug resistance and toxic side effects [16,17]. Resistance can occur when cancer cells are exposed to cisplatin for a prolonged period of time by increasing the expression of MDR-related proteins that release drugs out of the cell [18,19]. It has also been reported that EMT markers increase in some resistant cells [12,20]. Cisplatin-resistant cell lines could help in the generation of secondary treatments to overcome acquired drug resistance. However, few studies have been performed to establish a cisplatin-resistant model in oral cancer [21,22]. Our goal in this study was to establish and characterize three in vitro cisplatin-resistant cell models to provide the basis for studying the cisplatin resistance mechanism of the OSCC cell lines.

To study the mechanism of MDR in tumor cells, we first generated three cisplatin-resistant cell lines using long-term cisplatin exposure. These three cell lines, YD-8/CIS, YD-9/CIS, and YD-38/CIS, are not only more resistant to cisplatin than their parental cells, but they also show cross-resistance to other chemotherapeutic drugs. In MTT assays, YD-9/CIS cells (but not YD-8/CIS or YD-38/CIS cells) showed resistance to paclitaxel. In addition, YD-8/CIS, YD-9/CIS, and YD-38/CIS showed low, medium, and no resistance to CKD-602, respectively. None of the cell lines acquired a resistance to 5-fluorouracil. Therefore, our results show that the three cisplatin-resistant cell lines acquired resistance to cisplatin, and the cross-resistance to different chemotherapeutic drugs was cell-specific.

One of the common mechanisms of drug resistance is through a reduction in their intracellular accumulation, which is associated with overexpression of ABC transporters [23,24]. Thus, we compared the expression of the ABC transporter genes between the parental and cisplatin-resistant cells by qPCR. *BCRP* expression was significantly upregulated in YD-8/CIS and YD-38/CIS cells, and *MDR1* expression was significantly upregulated in YD-9/CIS cells; these results were confirmed via Western blotting. Therefore, it has been confirmed that long-term exposure of OSCC cell lines to cisplatin increased the expression of the ABC transporter genes, *MDR1* and *BCRP*. MDR1 is a 170 kDa ATP-dependent membrane transporter that is widely distributed throughout the body and pumps many foreign substances out of cells [25]. BCRP is a 72 kDa ATP-dependent membrane transporter which is localized apically in polarized cells to protect against xenobiotics, such as toxins or drugs [25].

Next, we compared the functional activities of MDR1 and BCRP between the parental and cisplatin-resistant cells using rhodamine 123 and bodipy FL prazosin accumulation assays. Rhodamine 123, a cationic fluorescent dye, is a good MDR1 substrate. Bodipy FL prazosin, a fluorescent derivative of the quinazoline based α-adrenergic antagonist prazosin, is known as a BCRP substrate [9,26,27,28]. Rhodamine 123 had a high level of efflux in the MDR1-overexpressing YD-9/CIS cells compared to parental cells, but not in YD-8/CIS and YD-38/CIS cells. The intracellular accumulation of bodipy FL prazosin was markedly reduced not only in BCRP-overexpressing YD-9/CIS, but also in YD-8/CIS and YD-38/CIS cells compared to parental cells, suggesting that prazosin is also transported by MDR1 [29,30]. Our results demonstrate that cisplatin-resistant OSCC cells increase the activity of MDR1 or BCRP proteins, causing drug resistance by reducing the intracellular accumulation of drugs.

Several studies have shown that MDR is associated with EMT [15,31]. Thus, we observed the morphology of the cells before and after acquired cisplatin-resistance. The cisplatin-resistant cells were more spindle-shaped than the round parental cells. We also found that the expression of N-cadherin was increased, while the expression of E-cadherin was decreased in the cisplatin-resistant cells. In addition, the cisplatin-resistant cells showed more migration in a wound healing assay than their parental cells. Therefore, our data show that these resistant cells had properties of EMT as well as drug resistance.

## 4. Materials and Methods

### 4.1. Reagents

Cisplatin (PubChem CID: 84691) was purchased from Choongwae Pharm. C. (Korea) and CKD-602 (PubChem CID: 123773277) was kindly provided by Chongkundang Pharm. Co. (Cheonan, Korea). These drugs were dissolved in distilled water at 1 mg/mL. 5-Fluorouracil (PubChem CID: 3385) was obtained from Sigma (St. Louis, MO, USA) and dissolved in DMSO at 20 mg/mL. Paclitaxel (PubChem CID: 36314) was purchased from Hanmi Pharm. Co. (Seoul, Korea). Each aliquot of the stock solutions was stored at −20 °C until use.

### 4.2. Cell Lines and Cell Cultures

Parental OSCC cell lines (YD-8, YD-9, and YD-38) were obtained from the Korean Cell Line Bank (Seoul, Korea), and cisplatin-resistant cell lines (YD-8/CIS, YD-9/CIS, and YD-38/CIS) were generated in our lab by exposing the original cell lines to increasing concentrations of cisplatin over a period of one year. The cell lines were maintained in RPMI-1640 medium (Biowest, Nuaillé, France) supplemented with 10% heat-inactivated FBS (Biowest, Nuaillé, France) and 1% penicillin-streptomycin (Biowest, Nuaillé, France) at 37 °C in a humidified atmosphere containing 5% CO_2_. When culturing the resistant cell lines, the final concentration of cisplatin was maintained at 2 μg/mL. 

### 4.3. Cell Viability Assays

For cell viability assays, single cell suspensions were seeded at 1 × 10^4^ cells per well in 96-well plates and then treated with cisplatin, 5-fluorouracil, paclitaxel, or CKD-602. Cells were incubated with the drugs for 72 h, after which 0.5 mg/mL of MTT reagent (Sigma, St. Louis, MO, USA) was added to each well. After 2 h, the medium was discarded, the resulting formazan was dissolved in DMSO, and the absorbance at 570 nm was measured using a SpectraMax Plus 384 microplate reader (Molecular Devices, Toronto, Canada). 

### 4.4. RNA Isolation and Real-Time qPCR

Total RNA was extracted from the harvested culture cells in 1.5 mL tubes using an RNA-spin Total RNA Extraction Kit (iNtRON Biotechnology Inc., Seongnam, Korea). cDNA synthesis was performed using High-Capacity cDNA Reverse Transcription Kits (Thermo Fisher Scientific, Waltham, Massachusetts, USA). The expression of *MDR1*, *MRP1-6*, *MRP9*, and *BCRP* was analyzed by a qPCR ViiA7 Real-Time PCR System using PowerUp SYBR Green Master Mix (Thermo Fisher Scientific, Waltham, Massachusetts, USA). The following primers were used: *MDR1*, 5′-CACCCGACTTACAGATGATG-3′ (forward) and 5′- GTTGCCATTGACTGAAAGAA-3′ (reverse); *MRP1*, 5′-GCCGAAGGAGAGATCATC-3′ (forward) and 5′-AACCCGAAAACAAAACAGG-3′ (reverse); *MRP2*, 5′-CGACCCTTTCAACAACTACTC-3′ (forward) and 5′-CACCAGCCTCTGTCACTTC-3′ (reverse); *MRP3*, 5′-GTGGGGATCAGACAGAGAT-3′ (forward) and 5′-TATCGGCATCACTGTAAACA-3′ (reverse); *MRP4*, 5′-GCTCAGGTTGCCTATGTGCT-3′ (forward) and 5′- CGGTTACATTTCCTCCTCCA-3′ (reverse); *MRP5*, 5′-CAGCCAGTCCTCACATCA-3′ (forward) and 5′-GAAGCCCTCTTGTCTTTTTT-3′ (reverse); *MRP6*, 5′-AGGAGGCCCGAGCTTAGAC-3′ (forward) and 5′-CCTGCCGTATTGGATGCTGT-3′ (reverse); *MRP9*, 5′-ATGCGGTTGTCACTGAAG-3′ (forward) and 5′-GTTGCCTCATCCATAATAAGAAT-3′ (reverse); *BCRP*, 5′-TGACGGTGAGAGAAAACTTAC-3′ (forward) and 5′-TGCCACTTTATCCAGACCT-3′ (reverse); and *GAPDH*, 5′-AATCCCATCACCATCTTCCA-3′ (forward) and 5′-TGGACTCCACGACGTACTCA-3′ (reverse) [32]. The relative expression of genes was calculated by the 2^−ΔΔCT^ method [33]. 

### 4.5. Western Blot Assay

To investigate the expression levels of multidrug resistant- or EMT-related proteins in the parent and resistant cell lines, we performed a Western blot assay. Whole cell lysates (20 μg) were resolved by 10% SDS-PAGE, and proteins were transferred onto a PVDF membrane (Bio-rad, Herculus, CA, USA). The membrane was incubated overnight at 4 °C with the following antibodies: anti-MDR1 from Abcam (Cambridge, UK), anti-BCRP from Santa Cruz Biotechnology (Santa Cruz, CA, USA), anti-E-cadherin and -N-cadherin purchased from CellSignaling Technology (Danvers, MA, USA), and anti-β-actin obtained from Sigma (St. Louis, MO, USA). The immunobands were detected using a ChemiDoc Touch imaging system (Bio-Rad, Hercules, CA, USA).

### 4.6. Rhodamine 123 and Bodipy FL Prazosin Accumulation Assays 

For the rhodamine 123 and bodipy FL prazosin accumulation assays, cells were seeded at 2 × 10^5^ cells per well in a 6-well plate. The next day, cells were treated with rhodamine 123 (1 μg/mL) or prazosin (250 nM) for 30 min. After incubation, cells were washed twice in ice-cold PBS, harvested by centrifugation at 1000 rpm for 5 min, and resuspended in 500 μL of PBS. Accumulation of rhodamine 123 or prazosin in cells was measured using a FACSCalibur flow cytometer (Becton–Dickinson, San Jose, CA, USA). At least 10,000 events were analyzed for each tube.

### 4.7. Immunofluorescence Analysis

E-cadherin and N-cadherin were analyzed by immunofluorescence in parental and cisplatin-resistant cells. Cells were seeded on cover slips in 6-well plates and incubated overnight at 37 °C. The next day, the cells were washed three times with PBS, fixed with 4% paraformaldehyde for 15 min and permeabilized with 0.1% NP-40 in PBS for 10 min at room temperature. The cells were blocked with 10% FBS in PBS for 30 min and incubated with either anti-E-cadherin or N-cadherin primary antibody (1:50 dilution) overnight at 4 °C. After this, the cells were washed three times with ice-cold PBS and incubated with Alexa Fluor_488 goat anti-rabbit secondary antibody (1:50 dilution) for 2 h at room temperature. The stained cells were then washed three times with PBS, mounted on slides using fluoroshield mounting medium with DAPI (ImmunoBioScience Corp., WA, USA) and observed with an AxioVision 4 imaging system (Zeiss, Oberkochen, Germany). 

### 4.8. Wound Healing Assay

For the wound healing assays, the cells (3 × 10^5^/well) were seeded into a 48-well plate and incubated for 24 h. Confluent cell cultures were scratched using a sterile pipette tip and then washed with PBS. After incubation for 8 h, the cells which migrated into the wound region were counted and captured using an Olympus CKX53 inverted microscope (Olympus Corporation, Tokyo, Japan).

### 4.9. Statistical Analysis

All the data were presented as the mean ± SD of at least three independent experiments. The significance of the difference between the two paired groups was analyzed using a Student’s *t-*test done in Microsoft Excel (Microsoft Corporation, Redmond, WA, USA). A *p* value of less than 0.01 was considered to be statistically significant.

## 5. Conclusions

In conclusion, our results show that we have successfully established three cisplatin-resistant human OSCC cell lines: YD-8/CIS, YD-9/CIS, and YD-38/CIS. These cell lines have highly similar biological characteristics, such as acquired drug resistance and EMT induction, and could be used as ideal models for further studies on the reversal of cisplatin-resistance in OSCC. They can also be used to help clarify the mechanism behind the molecular changes occurring in chemoresistance.

## Figures and Tables

**Figure 1 ijms-20-03034-f001:**
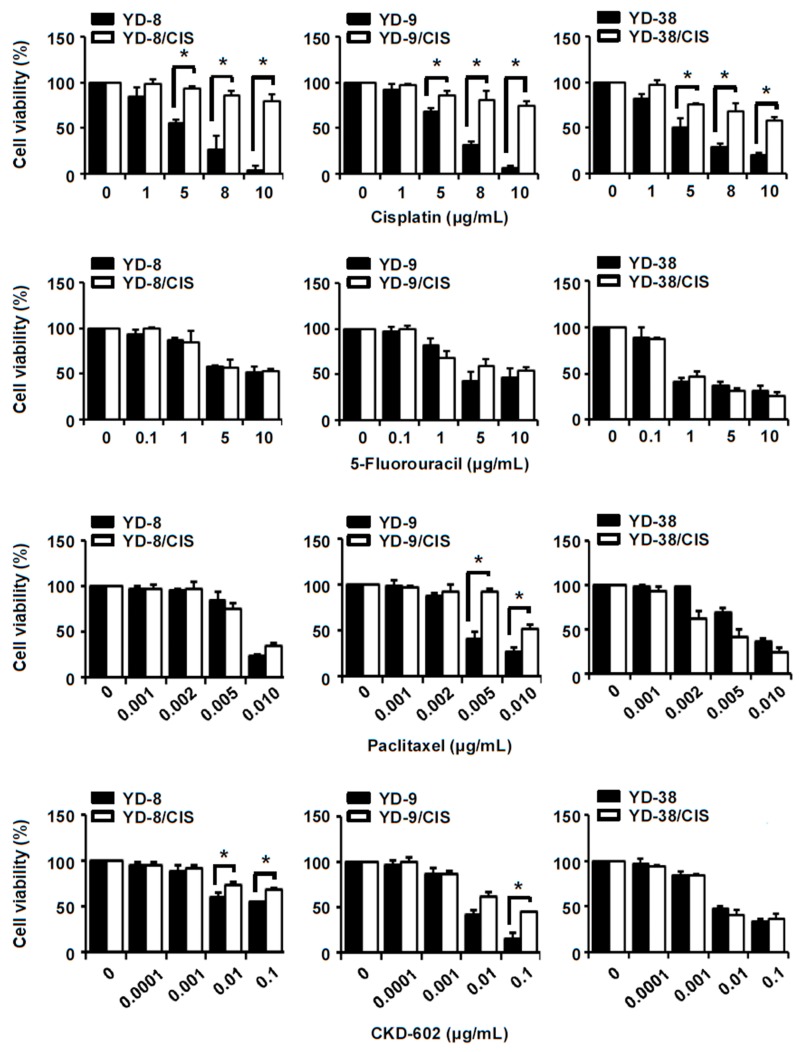
The effect of anticancer drugs on the cell viability of parental and cisplatin-resistant oral squamous cell carcinoma (OSCC) cell lines. YD-8, YD-8/CIS, YD-9, YD-9/CIS, YD-38, and YD-38/CIS cells were treated with cisplatin (0–10 μg/mL), 5-fluorouracil (0–10 μg/mL), paclitaxel (0–0.01 μg/mL), or CKD-602 (0–0.1 μg/mL) for 72 h. Cell viability was evaluated by MTT assay (mean ± SD; *n* = 5). Cisplatin-resistance was observed in YD-8/CIS, YD-9/CIS, and YD-38/CIS cells. Resistance to 5-fluorouracil, paclitaxel, and CKD-602 was not obtained in YD-8 and YD-38 cells. YD-9/CIS cells only showed cross-resistance to paclitaxel and CKD-602. * *p* < 0.01 versus parental cells.

**Figure 2 ijms-20-03034-f002:**
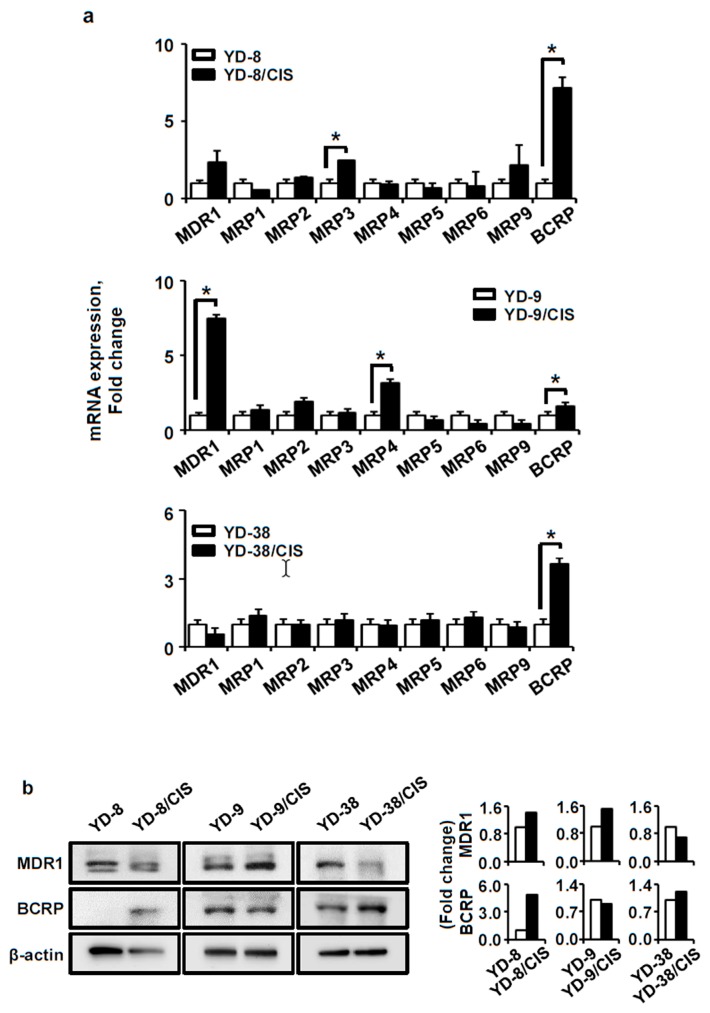
Differences in expression of multi-drug resistant (MDR)-related genes and proteins in parental and cisplatin-resistant OSCC cell lines. (**a**) The expression of MDR-related genes was measured by qPCR (mean ± SD; *n* = 3). * *p* < 0.01 versus parental cells. (**b**) Left, the expression of MDR1 and BCRP proteins was determined by a Western blot assay. Right, data represent quantitative results for left panel. Exposure to cisplatin induces upregulation of BCRP in YD-8 and YD-38 cells, and upregulation of MDR1 in YD-8 and YD-9 cells.

**Figure 3 ijms-20-03034-f003:**
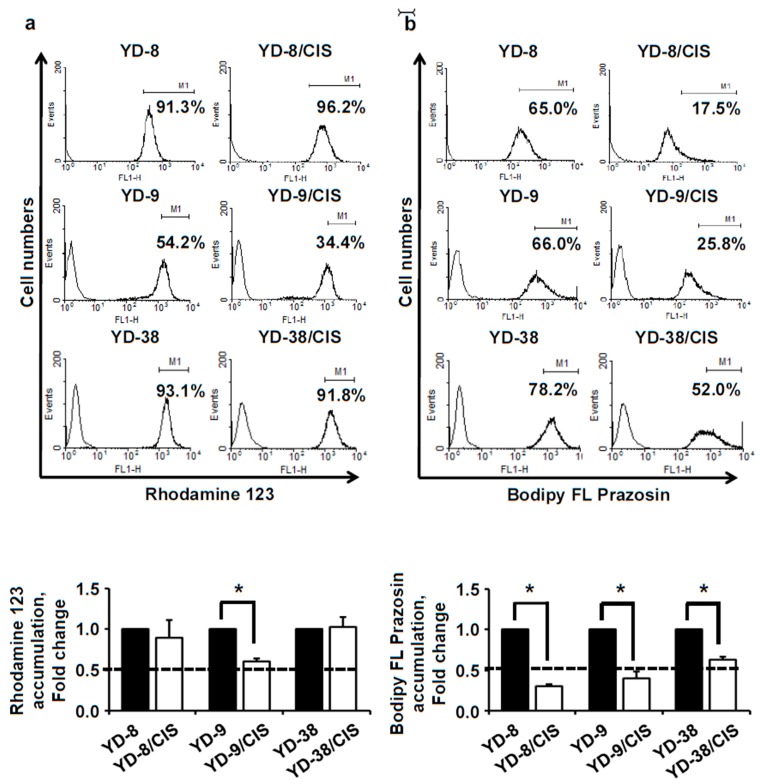
Functional assay for multidrug resistance protein1 or P glycoprotein (MDR1) and breast cancer resistance protein (BCRP) expression in parental and cisplatin-resistant OSCC cell lines. (**a**) Top, a rhodamine123 assay was used to monitor MDR1 activity. Bottom, data represent the quantitative results for the top panel. (**b**) Top, a prazosin assay was used to monitor BCRP activity. Bottom, data represent the quantitative results for the top panel. Cells were incubated for 30 min with growth medium alone (blank, thin solid line), or 1 μg/mL rhodamine123, or 250 nM prazosin (heavy solid line). Cisplatin treatment is sufficient to induce MDR1 activity in YD-9 cells and BCRP activity in YD-8, YD-9, and YD-38 cells (mean ± SD; *n* = 3). * *p* < 0.01 versus parental cells.

**Figure 4 ijms-20-03034-f004:**
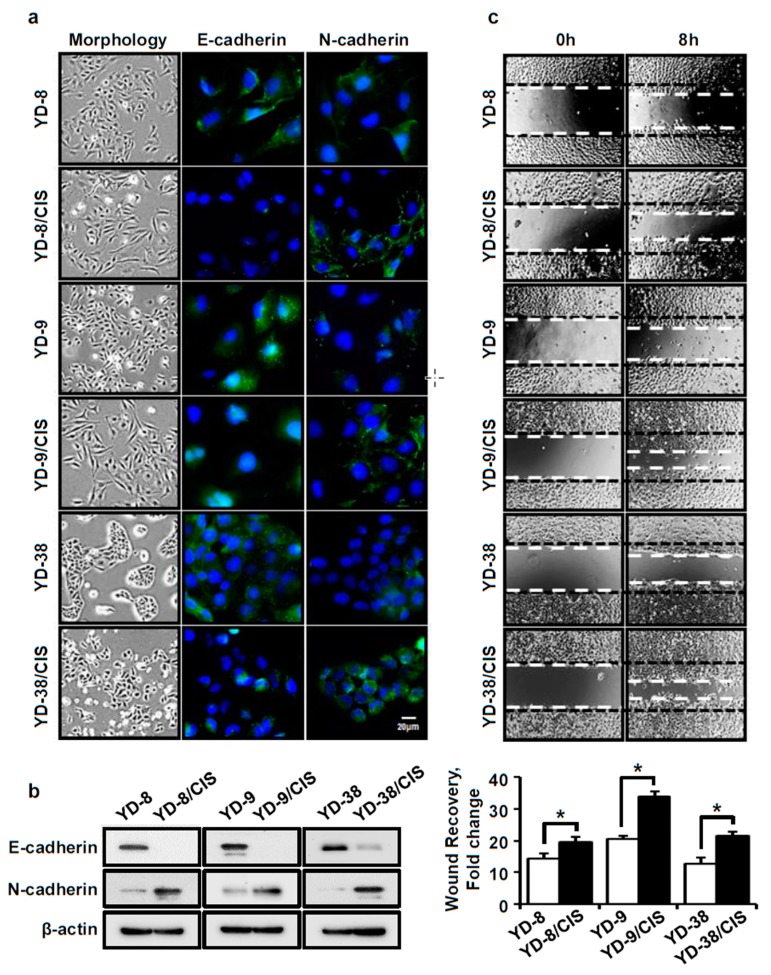
EMT-related markers in parental and cisplatin-resistant OSCC cell lines. (**a**) The morphological changes of cells were photographed using an Olympus CKX53 inverted microscope (×40). The expression levels of E-cadherin and N-cadherin (green signal) were determined by immunofluorescence assay. Blue signal corresponds to DAPI stained nuclei. Scale bar, 20 μm. (**b**) Western blotting confirmed the expression levels of E-cadherin and N-cadherin proteins. Cell mobility and N-cadherin expression increased in cisplatin-resistant cells compared to parental cells. (**c**) Top, cell migration was analyzed by a wound healing assay (×40). Bottom, data represent the quantitative results for the top panel (mean ± SD; *n* = 3). * *p* < 0.01 versus parental cells.

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
