# Peer review of "Upregulation of MDR- and EMT-Related Molecules in Cisplatin-Resistant Human Oral Squamous Cell Carcinoma Cell Lines"

_ijms, 2019, doi:10.3390/ijms20123034_

Round 1
Reviewer 1 Report
The authors have demonstrated the upregulation of MDR1 and BCRP proteins and EMT-related molecules in three cisplatin-resistant oral squamous cell carcinoma cell lines in their study “Upregulation of MDR- and EMT-related molecules in cisplatin-resistant human oral squamous cell carcinoma cell lines”. Acquired resistance to chemotherapeutic drugs is one of the leading cause of failure of chemotherapeutic treatment. If mechanisms leading to this drug resistance are known, it may help in finding treatment to overcome this chemo-resistance.
Following are my comments:
1) For all the experiments performed, did you have a control cell line, for example normal epithelial cells?
2) Line 48, Results section, for all experiments the authors should also discuss the concentration of drug used. For example, significant difference in viability of YD-9/CIS cells was observed with 5, 8 and 10 mg/ml of cisplatin.
3) Line 59, Results section, “Moreover, YD-9/CIS……………..CKD-602”. Similar to last comment, YD-9/CIS cells demonstrated cross-resistance to paclitaxel and CKD-602 at higher drug concentrations (0.005, 0.01 mg/ml and 0.1 mg/ml respectively)
4) Line 75-76, “BCRP was upregulated……………MDR1 was downregulated”. MDR1 was downregulated only in YD-38/CIS cells, not in YD-8/CIS.
5) Figure 2b, please quantify the protein expression from the band intensity of western blots.
6) Line 124, Discussion, please reference the line “However, few studies…………..model in oral cancer”.
7) Line 231, Immunofluorescence analysis, “cells were washed three times with ice-cold PBS”. In immunofluorescence assay, because you will be analyzing the cell structure you do not want the cells to change their morphology. If you use ice-cold PBS before fixation, morphology of cells tends to alter. Could you please explain why cells were washed with ice-cold PBS before fixing them.
Author Response
1) For all the experiments performed, did you have a control cell line, for example normal epithelial cells?
Thanks for your comments. The purpose of this experiment was to determine the difference between parental and resistant cells, so we couldn’t identify the normal epithelial cells. In the further experiment, we will consider your advice.
2) Line 48, Results section, for all experiments the authors should also discuss the concentration of drug used. For example, significant difference in viability of YD-9/CIS cells was observed with 5, 8 and 10 mg/ml of cisplatin.
Thanks for your comments. We rewrote the manuscript following your advices. "Figure 1 shows that YD-8/CIS, YD-9/CIS, and YD-38/CIS cells were less sensitive to cisplatin in a dose-dependent manner than their respective parental cells."
3) Line 59, Results section, “Moreover, YD-9/CIS……………..CKD-602”. Similar to last comment, YD-9/CIS cells demonstrated cross-resistance to paclitaxel and CKD-602 at higher drug concentrations (0.005, 0.01 mg/ml and 0.1 mg/ml respectively)
Thanks for your comments. We rewrote the manuscript following your advices. "Moreover, YD-9/CIS cells showed cross-resistance to paclitaxel and CKD-602 at higher drug concentrations (0.005, 0.01 μg/mL and 0.1 μg/mL respectively)."
4) Line 75-76, “BCRP was upregulated……………MDR1 was downregulated”. MDR1 was downregulated only in YD-38/CIS cells, not in YD-8/CIS.
We appreciate your comment. It was very helpful comments. We noticed something wrong with your advices and corrected the manuscript. "BCRP was upregulated in YD-8/CIS and YD-38/CIS cells, while MDR1 was downregulated only in YD-38/CIS cells. Conversely, MDR1 was overexpressed in YD-8/CIS and YD-9/CIS cells."
5) Figure 2b, please quantify the protein expression from the band intensity of western blots.
Thanks for your comments. We added a graph that quantify the protein expression from the band intensity of western blots.
6) Line 124, Discussion, please reference the line “However, few studies…………..model in oral cancer”.
Thanks for your comments. We added new references.
7) Line 231, Immunofluorescence analysis, “cells were washed three times with ice-cold PBS”. In immunofluorescence assay, because you will be analyzing the cell structure you do not want the cells to change their morphology. If you use ice-cold PBS before fixation, morphology of cells tends to alter. Could you please explain why cells were washed with ice-cold PBS before fixing them.
Thanks for your comments. We noticed something wrong with your advices and corrected the manuscript. We used PBS at room temperature before fixing the cells, and then ice-cold PBS was used.
Reviewer 2 Report
The manuscript entitled “Upregulation of MDR and EMT-related molecules in cisplatin-resistant human oral squamous cell carcinoma cell lines” reports generation of three cisplatin resistant cell lines in vitro by long-term exposure to cisplatin. The authors reported the changes in gene expression in drug exposed cell lines were associated with multi-drug resistance related genes and validated their qRT-PCR findings by protein measurements. They have evaluated cross-resistance of these cell lines and identified cell specific resistance against other chemotherapeutics. Authors also demonstrated that the drug resistant cell lines acquired EMT phenotype and showed increased cell migration potential. In this study, the gene expression analysis was restricted only to MDR related genes by qPCR methods and the study would have been comprehensive if the authors were using global gene expression analysis to identify genes associated with acquired drug resistance. Gene expression analysis between the resistant cell types may provide a better understanding about the cell specific cross resistance aspects. Overall, the study is interesting, and the paper is presented systematically. The reviewer feels the manuscript is appropriate for publication after the minor revisions mentioned above.
Author Response
The manuscript entitled “Upregulation of MDR and EMT-related molecules in cisplatin-resistant human oral squamous cell carcinoma cell lines” reports generation of three cisplatin resistant cell lines in vitro by long-term exposure to cisplatin. The authors reported the changes in gene expression in drug exposed cell lines were associated with multi-drug resistance related genes and validated their qRT-PCR findings by protein measurements. They have evaluated cross-resistance of these cell lines and identified cell specific resistance against other chemotherapeutics. Authors also demonstrated that the drug resistant cell lines acquired EMT phenotype and showed increased cell migration potential. In this study, the gene expression analysis was restricted only to MDR related genes by qPCR methods and the study would have been comprehensive if the authors were using global gene expression analysis to identify genes associated with acquired drug resistance. Gene expression analysis between the resistant cell types may provide a better understanding about the cell specific cross resistance aspects. Overall, the study is interesting, and the paper is presented systematically. The reviewer feels the manuscript is appropriate for publication after the minor revisions mentioned above.
Thank you for your good review comment, but this study is basically the main purpose of constructing resistant cell line in oral cancer cell line and basic resistance markers of established resistant cell line and to use it for future research. In the pilot study using RNA-seq, various resistance-related biomarker candidates are being analyzed and will be reported soon.
Reviewer 3 Report
Dear authors
in the paper three novel cisplatin-resistant OSCC cell lines were characterized. Although the paper is clearly presented, the novelty is not sufficient for publication. More mechanistic data are necessay.
Author Response
In the paper three novel cisplatin-resistant OSCC cell lines were characterized. Although the paper is clearly presented, the novelty is not sufficient for publication. More mechanistic data are necessary.
Thanks for your good review comments. Although we acknowledge the lack of novelty, this study is basically aimed at constructing resistant cell lines in oral cancer cell lines and to verify the basic resistance-related markers of established resistant cell lines and to use them for future research. As a follow-up study, we are analyzing various resistance-related biomarker candidates using RNA-seq and will report the findings as soon as possible.